# CAUSALLY FAIR NODE CLASSIFICATION ON NON-IID GRAPH DATA

## ABSTRACT

Fair machine learning seeks to identify and mitigate biases in predictions against unfavorable populations characterized by demographic attributes, such as race and gender. Recent research has extended fairness to graph data, such as social networks, but many neglect the causal relationships among data instances. This paper addresses the prevalent challenge in fair ML algorithms, which typically assume Independent and Identically Distributed (IID) data, from the causal perspective. We base our research on the Network Structural Causal Model (NSCM) framework and develop a Message Passing Variational Autoencoder for Causal Inference (MPVA) framework to compute interventional distributions and facilitate causally fair node classification through estimated interventional distributions. Theoretical soundness of the proposed method is established under two general and practical conditions: Decomposability and Graph Independence. These conditions formalize when interventional distributions can be computed using *do*-calculus in non-IID settings, thereby grounding the framework in rigirous causal inference theory rather than imposing ad hoc constraints. Empirical evaluations on semi-synthetic and real-world datasets demonstrate that MPVA outperforms conventional methods by effectively approximating interventional distributions and mitigating bias. The implications of our findings underscore the potential of causality-based fairness in complex ML applications, setting the stage for further research into relaxing the initial assumptions to enhance model fairness.

## 1 INTRODUCTION

Amid the increasing prevalence of machine learning algorithms and models in the real world, ensuring unbiased predictions by identifying and mitigating biases is essential to maintaining equity and promoting the reliability of deploying machine learning to high-stake scenarios (Caton & Haas, 2020; Zafar et al., 2017a;b; Mehrabi et al., 2021; Pessach & Shmueli, 2023; Zliobaite, 2017; Quy et al., 2022; Wan et al., 2022). The past decade has seen the development of numerous fair learning algorithms designed to build fair machine-learning systems, which are grounded in various notions of fairness, including both statistical-based and causality-based ones (Pedreschi et al., 2009; Hardt et al., 2016; Zhang & Bareinboim, 2018b; Wen et al., 2019).

Despite significant progress in fair machine learning, many existing algorithms rely on the assumption of Independent and Identically Distributed (IID) (Caton & Haas, 2020; Zafar et al., 2017a; Hardt et al., 2016; Zafar et al., 2017b). In real-world scenarios, however, data instances are seldom independent and often exhibit connections, which are referred to as non-IID settings[1]. For instance, in predicting loan defaults, while each individual's data point appears independent, an individual's loan repayment behavior can be influenced by the experiences of their friends and family. To address the above issues, the research community has extended the fairness notions and studied fair machine learning in the context of graph mining. Group fairness notions like demographic parity, equalized odds, equal opportunity, and so on, have been extended to graph machine learning settings (Dong et al., 2022; Fan et al., 2021; Bose & Hamilton, 2019; Zhu et al., 2024b; Luo et al., 2024), and new fairness notions such as degree-related fairness and node pair distance-based fairness (Kang et al., 2020; Dong et al., 2021) have also been proposed.

---

[1] In this paper, the terms 'non-IID settings', 'graph settings', and 'network settings' are used interchangeably and referred to the situations where data instances are interconnected.

However, a significant gap in the current literature on fair graph learning is the lack of principled exploration of causality-based methods. Causal fairness plays a vital role in the fair machine learning field by modeling unfairness as the causal effect of the sensitive feature on the model prediction rather than relying solely on correlation. While causality-based fairness has been extensively studied in IID settings, with notions such as direct and indirect discrimination, counterfactual fairness, and path-specific counterfactual fairness being well-established (Zhang & Bareinboim, 2018a; Chiappa, 2019; Malinsky et al., 2019), its application in non-IID graph settings remains largely underexplored, despite that recent studies have highlighted that directly applying conventional fairness notions without accounting for dependencies among individuals can yield biased outcomes (Zhang et al., 2022; Zhang, 2023). One key reason for this gap is that established causal inference frameworks, such as structural causal models (SCMs) and *do*-calculus, typically assume IID conditions. Although causal inference for non-IID settings has been studied in the context of interference (Hudgens & Halloran, 2008; Tchetgen & VanderWeele, 2012), the integration of these methods into machine learning workflows to measure causal unfairness in non-IID graph data presents considerable challenges, due to computation and estimation barriers such as the consistent interference assumption (Arbour et al., 2016a; Lee & Honavar, 2016; Arbour et al., 2016b). While preliminary efforts have explored formulating causal fairness in non-IID settings, e.g., (Agarwal et al., 2022; 2021; Ma et al., 2022; Yang et al., 2024), there studies lack a rigorous foundation for extending traditional inference techniques, such as *do*-calculus, to graph-structured data. A comprehensive theoretical and generalized framework for causal inference in such settings is still lacking.

In this work, we develop a foundational framework for causality-based fair graph machine learning. We base our research on the graphical extension of the SCM, known as the Network Structural Causal Model (NSCM) (Ogburn et al., 2022). To conduct causal inference in graph settings, we first identify conditions on the data generation mechanism that permit causal inference in the NSCM by using the *do*-calculus, and then based on which we derive theoretically sound approaches for computing interventional distributions given observational data. Specifically, we leverage the Weisfeiler-Lehman (WL) algorithm and use the node color to represent the structural information with each node. In the graph settings, the structural information of each node are unique, leading to unique values for the node color and posing computational challenges for causal inference. To address this challenge, we introduce two general conditions that ensure the theoretical soundness of our framework, namely *Decomposability*, which posits that the structural equation in the NSCM can be decomposed into a message-passing component and a transformation component, and *Graph Independence* which posits that the node color is independent with the outcome variable of the causal inference. We show that, under the above two conditions, the interventional distribution of the outcome variable can be readily computed from the observational graph data using the classic *do*-calculus. Based on the connection between the WL algorithm and message-passing neural networks (MPNN), we develop a deep neural net-based implementation of our causal inference approach, named Message Passing Variational Autoencoder for Causal Inference (MPVA), which combines the MPNN with a conditional variational autoencoder (cVAE).

To address fair node classification, we first train the MPVA independently to infer causal effects. We then integrate the MPVA with a node classifier to formulate a causal fairness regularization term. The resulting combined model can be trained using standard learning algorithms, producing a node classifier that incorporates causal fairness. We conduct empirical evaluations of our fair node classification method on both semi-synthetic and real-world graph datasets. The results show that the proposed MPVA effectively approximates the interventional distribution and accurately estimates causal effects in non-IID graph settings. Building on this capability, our bias mitigation algorithm achieves fair node classification, outperforming baseline methods that fail to address these challenges.

## 2 PRELIMINARY

**Structural Causal Model (SCM)**. Structural causal models (Pearl, 2009) provide a mathematical framework to understand causal relationships within a system. SCMs define the causal dynamics of a system through a collection of structural equations. Each variable $X$ in the system is associated with a function $f_X$, such that $x = f_X(\mathsf{pa}_X, u_X)$. Here, $\mathsf{pa}_X$ denotes the values of other endogenous variables that directly influence $X$, and $u_X$ represents the values of the exogenous variables impacting $X$. Each SCM is associated with a causal diagram consisting of a set of nodes for representing

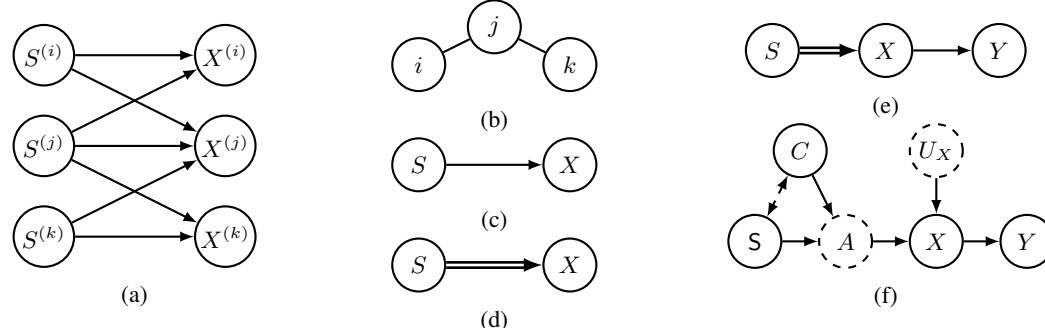

Figure 1: Graphs and diagrams. (a) The interference graph. (b) The network $\mathcal{G}$. (c) The causal diagram $\mathcal{C}$. (d) The networked causal diagram $\mathcal{N}$. (e) The networked causal diagram for node classification. (f) The causal diagram that is equivalent to the networked causal diagram in Fig. 1e.

variables and a set of directed edges for representing the direct causal relations. We posit the causal Markovian model in this paper, i.e., all exogenous variables are mutually independent.

**Causality-based Fairness Notions**. Defining causality-based fairness notions is facilitated with the *do*-operator (Pearl, 2009), which simulates the physical interventions that force some variable to take certain values. Formally, the intervention that sets the value of $S$, a sensitive demographic characteristic (i.e., race or gender), to $s$ is denoted by $do(S = s)$. The distribution of a variable $X$ after the intervention on $s$ is called the interventional distribution, denoted as $P(x|do(s)) := P(x|do(S = s))$. Causality-based fairness notions are usually defined by the disparity in the interventional distribution across different demographic groups, such as the total effect (Zhang & Bareinboim, 2018a), direct discrimination (Zhang et al., 2017), indirect discrimination (Zhang et al., 2017), and counterfactual fairness (Kusner et al., 2017). In this paper, we consider the total effect of $S$ on $Y$ defined as $\mathbb{E}[Y|do(s^+)] - \mathbb{E}[Y|do(s^-)]$ where $s^+$ and $s^-$ represent the favorable and unfavorable groups of the demographic characteristics.

## 3 PROBLEM FORMULATION

### 3.1 NETWORK STRUCTURAL CAUSAL MODEL (NSCM)

Traditional SCM assumes that data instances are IID. To deal with non-IID data, recent studies have proposed to extend the SCM to capture the interference between individuals (e.g., (Ogburn et al., 2022)). These extensions usually involve using interference graphs (as illustrated in Fig. 1a) to describe both the causal relationship between features and the interference relationship between individuals. However, previous studies (e.g., (Arbour et al., 2016a; Lee & Honavar, 2016; Arbour et al., 2016b)) usually make the consistent interference assumption, i.e., the interference graph is consistent among intersected instances, which is often violated in real-world scenarios. In our work, we adopt a more general framework extended from traditional SCM, referred to as the Network Structural Causal Model (NSCM). To distinguish the network for describing the existence of potential interference and the graph for describing the causal relationship between features, NSCM explicitly considers the following two components: 1) network ($\mathcal{G}$), where each node represents an individual or instance and the edges depict the potential for interference between connected individuals; and 2) causal diagram ($\mathcal{C}$), where each node represents a feature and the edges depict the parent-child relationship between features determined by the structural equations. It is formally defined as follows.

**Definition 1** (Network Structural Causal Model (NSCM))**.** An NSCM $\mathcal{M}$ is a quadruple $\mathcal{M} = \langle \mathcal{G}, \mathbf{U}, \mathbf{V}, \mathbf{F} \rangle$ where

1. $\mathcal{G}$ is a network that consists of a set of connected nodes.

2. $\mathbf{U}$ is a set of exogenous variables. For every node $i$ in the graph, $\mathbf{u}^{(i)} \in \mathbf{U}$ is an instantiation of the exogenous variables received by node $i$.

3. $\mathbf{V}$ is a set of endogenous variables. For every node $i$ in the graph, $\mathbf{v}^{(i)} \in \mathbf{V}$ is an instantiation of the endogenous variables received by node $i$.

4. $\mathbf{F}$ is a set of structural equations. For each variable $X \in \mathbf{V}$ and a node $i$, an equation $x^{(i)} = f(\mathsf{pa}_X^{(i)}, \{\mathsf{pa}_X^{(j)} : j \in \mathsf{ne}^{1:k}(i)\}, u_X^{(i)})$ determines the value of $x^{(i)}$, where $\mathsf{ne}^{1:k}(i)$ denotes the neighborhood of $i$ within $k$ hops.

An illustrative example of an NSCM with two variables $S, X$ is shown in Figs. 1a, 1b and 1c, which show the interference graph, the network $\mathcal{G}$, and the causal diagram $\mathcal{C}$. The structural equations of this NSCM can be given by

$$x^{(i)} = f(s^{(i)}, \{s^{(j)} : j \in \mathsf{ne}^{1:k}(i)\}, u_X^{(i)})$$

To further simplify representation, we introduce a hybrid graph, the networked causal diagram $\mathcal{N}$, which integrates network information with the causal diagram while omitting detailed interference structure, as shown in Fig. 1d. In the networked causal diagram, the solid line arrow represents the case where the causal effect is transmitted only from a variable of an individual to another variable of the same individual as seen in the traditional SCM (not showing in this example), while the double line arrow represents the existence of interference between different individuals in $\mathcal{G}$ when the causal effect is transmitted from one variable to another. For $k = 1$, the interference solely occurs between nodes that are immediate neighbors, while for $k > 1$, the interference propagates through multiple hops in a message-passing fashion. Given the NSCM formulation, the causal inference task is to infer the interventional distribution $P(x|do(s)) := P(x|do(S = s))$ from observational graph data.

### 3.2 Fair Node Classification

We will apply our causal inference techniques to the fair node classification problem. Denote the sensitive feature by $S$, the non-sensitive feature by $X$, and the decision by $Y$. We consider a general networked causal diagram as shown in Fig. 1e, where, for simplicity, we posit that interference only exists from $S$ to $X$. However, our methods can be applied to deal with the interference between any pair of variables. Suppose that we are given a dataset $\mathcal{D} = \{s^{(i)}, x^{(i)}, y^{(i)}\}_{i=1}^K$ and a network $\mathcal{N}$ that connects individuals in $\mathcal{D}$ to reflect interference. The goal is to build a classifier $h : X \mapsto Y$ for predicting the label. We say that the classifier is causally fair if $\mathbb{E}[h(x)|do(s^+)] = \mathbb{E}[h(x)|do(s^-)]$, where $do(\cdot)$ represents the intervention that is conducted under the networked causal diagram.

## 4 Method

### 4.1 Causal Inference on Network Data

We use the networked causal diagram in Fig. 1e and the task of inferring the causal effect from $S$ to $X$ (i.e., computing $P(x|do(s))$) as a running example for elaborating our method. The $do$-calculus is an axiomatic system that is widely used for solving the causal inference problem ((Pearl, 2009), Ch. 3.4). However, as shown in recent studies (Zhang et al., 2022; Zhang, 2023), directly applying $do$-calculus will lead to biased results if the IID assumption is not satisfied. This section presents a theoretical study aimed at extending the intervention $do$ operator and symbolic notations to NSCM, starting by identifying conditions that are more general and less restrictive than the consistent interference assumption.

Let $\mathsf{s}^{(i)} := \{s^{(i)}, \{s^{(j)} : j \in \mathsf{ne}^{1:k}(i)\}\}$ denote a multiset of all neighboring information of $S$ that affects node $i$ determined by $\mathsf{ne}^{1:k}(i)$ (including node $i$ itself). Then, let $do(\mathsf{s}^{(i)} = s)$ denote the intervention that changes the value of $S$ of every node in $\mathsf{s}^{(i)}$ to $s$, and $do(\mathsf{s}) = s$ means to perform $do(\mathsf{s}^{(i)} = s)$ for each $i$. Thus, in our context, we have that $P(x|do(\mathsf{s}) = s)$ is equivalent to $P(x|do(s))$ in the graph setting, which is the target quantity we aim to compute.

The key idea in computing the interventional distribution $P(x|do(\mathsf{s}) = s)$ is to decouple the causal effect from the influence of interference in the causal mechanism linking $S$ and $X$. Specifically, we first leverage the concept of node color in the Weisfeiler Leman (WL) graph isomorphism test to represent local graph structure information. The node color represents the structural information of each node. In the WL graph isomorphism test, two arbitrary nodes are assigned the same colors if

and only if they have the same local graph structures. Let $c^{(i)}$ be the color of node $i$ obtained by WL algorithm with identical initial node colors. As shown in the literature (Jegelka, 2022), we have:

**Lemma 1** ((Jegelka, 2022)). *For two different nodes $i, j$, $c^{(i)} = c^{(j)}$ if and only if nodes $i$ and $j$ have identical computation trees in the WL graph isomorphism test.*

Based on this concept, we propose two key conditions below, which will permit causal inference in NSCM by using the *do*-calculus.

**Condition 1** (Decomposability). The structural equation in NSCM can be decomposed into a message-passing mechanism that represents the aggregated causal effect of the neighborhood and an internal causal mechanism that represents the causal effect transmitted within the node.

The Decomposability condition allows us to create artificial variables to represent the aggregated influence from the neighborhood. Specifically, with this condition, the structural equation $x^{(i)} = f(\mathsf{s}^{(i)}, \{\mathsf{s}^{(j)} : j \in \mathsf{ne}^{1:k}(i)\}, u_X^{(i)})$ can be decomposed into two equations:

$$a^{(i)} = f^{MP}(\mathsf{s}^{(i)}), \qquad x^{(i)} = f^{INT}(a^{(i)}, u_X^{(i)})$$

where function $f^{MP}$ represents the message-passing mechanism based on the given graph, $a^{(i)}$ is an intermediate variable, and function $f^{INT}$ represents the internal causal mechanism within the node.

According to Lemma 1 and Condition 1, we have the following proposition, which allows a valid mapping $g : \{\mathsf{S}, C\} \mapsto A$ to represent the message-passing mechanism.

**Proposition 2.** *For two different nodes $i, j$, if $\{\mathsf{s}^{(i)}, c^{(i)}\} = \{\mathsf{s}^{(j)}, c^{(j)}\}$, then we have $a^{(i)} = a^{(j)}$.*

*Proof.* According to Lemma 1, nodes $i, j$ have identical computation trees. According to (Xu et al., 2019), any message passing network maps nodes $i, j$ to the same embedding. □

**Condition 2** (Graph Independence). Exogenous variable $U_X$ is independent of node color $C$.

The Graph Independence condition posits that the graph structure is independent of the exogenous variable, which ensures that all relevant information from the neighborhood regarding the interventional value of $X$ can be effectively summarized in the intermediate variable $A$.

As a result, the implication of Proposition 2 and Condition 2 together implies that we can convert the networked causal diagram in Fig. 1d to an equivalent causal diagram as shown in Fig. 1f, where $A$ is a latent variable and the dashed arrow between $\mathsf{S}$ and $C$ represents the possible hidden confounding due to the biased distribution of $S$ in the network. Since Fig. 1f is a traditional causal diagram, we can adopt the *do*-calculus to compute $P(x|do(\mathsf{s}) = s)$, as shown in Theorem 3 below.

**Theorem 3.** *Given the causal diagram in Fig. 1f, we have*

$$P(x|do(\mathsf{s}) = s) = \sum_c P(c) \sum_a P(x|a) P(g(\mathsf{s}, c) = a). \tag{1}$$

Please refer to the appendix for the proof. Theorem 3 suggests that $P(x|do(\mathsf{s}) = s)$ can be computed if we have observed the intermediate variable $A$, which is latent. However, according to the Hedge Criterion (Shpitser & Pearl, 2008), i.e., the graphical criterion of identifiability, $P(x|do(\mathsf{s}) = s)$ is identifiable even if $A$ is unobserved. In fact, note that for observational distribution $P(x)$, we have

$$P(x) = \sum_{\mathsf{s},c,a} P(\mathsf{s}, c) P(g(\mathsf{s}, c) = a) P(x|a) = \sum_{\mathsf{s},c} P(\mathsf{s}, c) \sum_a P(x|a) P(g(\mathsf{s}, c) = a). \tag{2}$$

By examining Eqs. 1 and 2 together, we see the two equations share the term $\sum_a P(x|a) P(g(\mathsf{s}, c) = a)$, which is independent of $A$. This suggests that we can first estimate $P(x)$, e.g., by fitting a machine learning model. Then, without needing to explicitly learn the true values of $A$, we can compute $P(x|do(\mathsf{s}) = s)$ by performing the intervention $do(\mathsf{s}) = s$ on the network and using the previously learned model for inference. Hence, we have the following corollary.

**Corollary 4.** *Given the causal diagram in Fig. 1f, $P(x \mid do(\mathsf{s}) = s)$ can be approximated using the same model employed for estimating $P(x)$, with the intervention $do(\mathsf{s}) = s$ applied to the network.*

This corollary motivates our Message Passing Variational Autoencoder for Causal Inference (MPVA) framework that combines a message-passing neural network (MPNN) with a conditional variational autoencoder (cVAE), as detailed in the next subsection.

## 4.2 MESSAGE PASSING VARIATIONAL AUTOENCODER FOR CAUSAL INFERENCE (MPVA)

Next, we develop the MPVA framework as motivated by the above analysis. For generalization, we further consider the existence of independent variables $Z$ other than $S$ that directly affect $X$, and Corollary 4 readily applies to this case. In this framework, we first use an MPNN to learn a hidden representation capturing the aggregated causal effect of $S$ on $X$ from each node's neighborhood, denoted as $\hat{a} = \hat{g}^{MPNN}(\mathsf{s})$. Then, we use the estimated intermediate representation $\hat{A}$ along with the variables $Z$ as inputs to a multilayer perceptron (MLP) to predict the node feature $X$, denoted as $\hat{x} = MLP(\hat{a}, z)$. After that, we use a cVAE for learning the conditional distribution $P(x|\hat{a}, z)$, where the encoder $v = EN(x, \hat{a}, z)$ takes $x$ as the input with $\hat{a}, z$ as conditions, and the decoder $\hat{x} = DE(v, \hat{a}, z)$ attempts to reconstruct $x$. The architecture of MPVA is illustrated in Fig. 2.

In the training phase, we first train MPNN and MLP with the dataset $\mathcal{D}$ and network $\mathcal{N}$. Note that in this process, we do not require actual values of $A$ as supervised signals. Then, we freeze the parameters of MPNN and MLP and train the cVAE. In the inference phase, for each node $i$, we first use the MPNN to compute $\hat{a}^{(i)} = \hat{g}^{MPNN}(\mathsf{s}^{(i)})$ and use the encoder of the cVAE to compute $v^{(i)} = EN(x^{(i)}, \hat{a}^{(i)}, z^{(i)})$. Then, we perform the intervention $do(\mathsf{s}) = s$ to change the value of $S$ to $s$ for all nodes. After that, we use the MPNN to compute the value of $A$ again under the intervention, i.e., $\tilde{a}_s = \hat{g}^{MPNN}(do(\mathsf{s}^{(i)}) = s)$, and use the cVAE to reconstruct $X$ using $\tilde{a}_s$ and $v^{(i)}$, i.e., $\tilde{x}_s^{(i)} = DE(v^{(i)}, \tilde{a}_s, z^{(i)})$. As a result, $\tilde{x}_s^{(i)}$ is the interventional value of $x_s^{(i)}$ under the intervention $do(\mathsf{s}) = s$. Note that in this inference process, our method leverages the Abduction-Action-Prediction framework which is well-known as the standard method for counterfactual inference. However, since our method computes interventional variants rather than counterfactual variants, it does not have identifiability issues associated with counterfactual estimation.

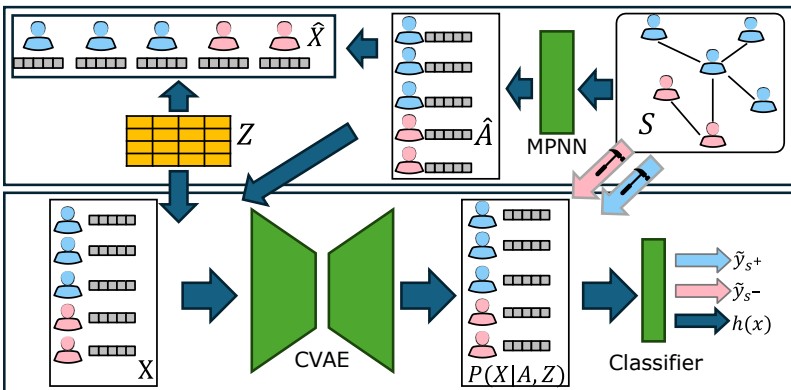

Figure 2: The MPVA framework. MPNN learns the hidden representation of aggregated causal effects from neighbors through reconstruction. cVAE learns the conditional distribution for computing the interventional distribution.

## 4.3 CAUSALLY FAIR NODE CLASSIFICATION

Based on the aforementioned causal effect estimation, we conceptualize causal fair node classification as a regularized optimization problem. To this end, we utilize the learned interventional distribution from cVAE to develop an additional penalization term, which is appended to the traditional classification loss. Specially, we generate the interventional variants using the classifier $h$ and the learned MPVA, denoted as $\tilde{y}_{s+}$ and $\tilde{y}_{s-}$, where $\tilde{y}_s = h(\tilde{x})|do(s)$. To address fairness in the classification task, we construct a regularization term aimed at minimizing the causal discrepancy between two interventional variants. This term is defined as follows:

$$\ell_f = \mathbb{E}[h(\tilde{x})|do(s^+)] - \mathbb{E}[h(\tilde{x})|do(s^-)]$$
$$= \mathbb{E}[\mathbb{1}_{\tilde{y}_{s+}=1}] - \mathbb{E}[\mathbb{1}_{\tilde{y}_{s-}=1}] = \mathbb{E}[\mathbb{1}_{\tilde{y}_{s+}=1}] + \mathbb{E}[\mathbb{1}_{\tilde{y}_{s-}=-1}] - 1, \tag{3}$$

where $\mathbb{1}$ is the indicator function. Following (Wu et al., 2019), the indicator function can be further replaced with the differentiable surrogate function $u(\cdot)$. It is noteworthy that this differentiability ensures that the regularization term can be effectively incorporated into the classic loss functions

used for training the node classifier. To sum up, this regularization term can be seamlessly incorporated with the classification loss: $\ell = \frac{1}{n}\Sigma_{i\in[K]}\ell_c(h(x^{(i)}), y^{(i)}) + \lambda\ell_f$, where $\ell_c$ is the empirical loss function and $\lambda$ is a hyper-parameter to balance model performance and causal fairness.

## 5 EXPERIMENT

We evaluate the proposed method and comparisons on both semi-synthetic and real-world graphs. The detailed statistics of these datasets are included in the appendix, including the number of nodes, the number of edges, and the dimension of features. The implementation details are also available in the appendix. The code is attached as a zip file in the Supplementary Material.

### 5.1 DATASETS

In our experiments, we adopt both semi-synthetic datasets, which allow full control over the data-generation process, and real-world datasets, which evaluate the external generalizability of our methods. Semi-synthetic data are commonly used in the causal inference and fair machine learning fields since the ground truth, e.g., causal effects and bias, cannot be directly observed in real-world settings. To create semi-synthetic datasets, we adapt the Credit Dataset (Yeh, 2016) by introducing network structures and causal relationships using the Network Structural Causal Model. This approach allows us to precisely control data generation and accurately derive ground-truth interventional distributions for arbitrary interventions on the sensitive attribute. Specifically, we generate two semi-synthetic datasets, denoted as D1 and D2, for evaluation purposes. We also conduct experiments on widely used real-world datasets, namely Credit Defaulter (Yeh, 2016) and German (Hofmann, 1994). Since the underlying mechanisms for the real-world dataset are unknown, we evaluate our proposed framework's ability to estimate non-IID causal interventions and compare it against baseline methods, including IID-based causal fairness approaches. We use the learned MPVA model to measure the interventional quantities and evaluate the performance in terms of non-IID causal fairness. Additional details on the datasets are provided in the appendix.

### 5.2 EXPERIMENT SETTINGS

**Fairness Metrics:** We evaluate the performance of the proposed framework in terms of prediction accuracy and fairness. For non-causal fairness notions, we use demographic parity, which is a widely used fairness notion in the fairness-aware learning field, to evaluate the fairness performance at the group level. Demographic parity requires the decision made by the classifier to be independent of the sensitive attribute. Usually, it is quantified with regard to *risk difference* (**RD**), i.e., the difference in the positive predictions between the sensitive group and the non-sensitive group. It can be expressed as $\left|\mathbb{E}_{X|S=s^+}[\hat{Y}] - \mathbb{E}_{X|S=s^-}[\hat{Y}]\right|$. For causal fairness notions, we consider both the IID and the non-IID (i.e., graph-based) causal fairness notions. We denote the IID causal fairness as **CF** whose calculation and estimation approaches are described in the appendix. On the other hand, we denote the graph-based causal fairness notion as **gCF**, which is described in Eq. equation 3.

**Mitigation Baselines:** We compare the proposed framework **MPVA** with several state-of-the-art non-IID bias mitigation methods and the conventional IID constraint-base methods. **GCN-RD** and **GCN-IID** are the conventional Graph Convolutional Networks (GCN) with the risk difference and the causal fairness constraints developed for the IID data. The constraint formulation is described in the appendix. **FairGNN** (Dai & Wang, 2021) employs a covariance-based adversarial discriminator to predict the sensitive attribute into the conventional GNN node classifier. **GEAR** (Ma et al., 2022) utilizes a variational auto-encoder to synthesize counterfactual samples to achieve counterfactual fairness for node classification. **NIFTY** (Agarwal et al., 2021) enhances fairness and stability in GNNs by introducing a novel objective function and layer-wise weight normalization based on the Lipschitz constant. **FairINV** (Zhu et al., 2024a) trains fair GNNs by eliminating spurious correlations between labels and sensitive attributes within a single training session. For the detailed implementation of baseline methods and sensitivity analysis, please refer to the implementation section in the appendix.

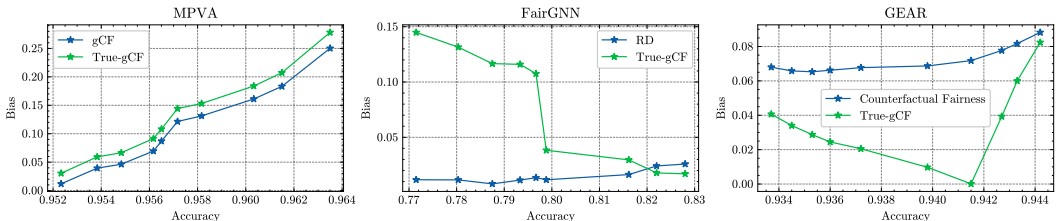

Figure 3: Comparison of measured bias and true gCF bias on D2 with various mitigation methods.

Table 1: Fairness measurement of conventional GCN using various metrics on semi-synthetic datasets.

| Data | Acc | RD | CF | gCF | True-gCF |
|---|---|---|---|---|---|
| Semi-synthetic D1 | $0.9674_{\pm 0.0017}$ | $0.0580_{\pm 0.0015}$ | $0.0319_{\pm 0.0001}$ | $0.1792_{\pm 0.0344}$ | $0.1960_{\pm 0.0338}$ |
| Semi-synthetic D2 | $0.9715_{\pm 0.0011}$ | $0.0832_{\pm 0.0013}$ | $0.0461_{\pm 0.0003}$ | $0.6721_{\pm 0.0132}$ | $0.6884_{\pm 0.0084}$ |

Table 2: Evaluation of mitigation methods on semi-synthetic datasets.

| Data | Metric | Own Metric | Acc | gCF | True-gCF |
|---|---|---|---|---|---|
| | MPVA | - | $0.9259_{\pm 0.0051}$ | $\mathbf{0.0023_{\pm 0.0012}}$ | $\mathbf{0.0010_{\pm 0.0006}}$ |
| Semi-synthetic D1 | GCN-RD | $0.0074_{\pm 0.0074}$ | $0.8743_{\pm 0.0179}$ | $0.4150_{\pm 0.0961}$ | $0.4219_{\pm 0.0969}$ |
| | GCN-IID | $0.0031_{\pm 0.0015}$ | $0.9024_{\pm 0.0071}$ | $0.1787_{\pm 0.0437}$ | $0.1896_{\pm 0.0456}$ |
| | MPVA | - | $0.9490_{\pm 0.0009}$ | $\mathbf{0.0019_{\pm 0.0018}}$ | $\mathbf{0.0073_{\pm 0.0048}}$ |
| Semi-synthetic D2 | GCN-RD | $0.0091_{\pm 0.0055}$ | $0.8828_{\pm 0.0026}$ | $0.1634_{\pm 0.0744}$ | $0.1868_{\pm 0.0900}$ |
| | GCN-IID | $0.0069_{\pm 0.0010}$ | $0.9094_{\pm 0.0021}$ | $0.6818_{\pm 0.0310}$ | $0.6977_{\pm 0.0292}$ |

## 5.3 RESULTS ON SEMI-SYNTHETIC DATA

We first generate two network structures with different generating parameters for the semi-synthetic datasets. To show the biases contained in the generated graphs in terms of the proposed metrics, as well as the effectiveness of MPVA for estimating gCF, we train the classic GCN models without any bias mitigation considerations. Given the conventional GCN node classification models, we measure the bias and report the results as shown in Tab. 1, which present the empirical node prediction accuracy, the estimated RD, CF, gCF (highlighted on blue), as well as the ground truth gCF (highlighted on green) that is directly computed by performing interventions on the true causal model. As we can see, the node prediction accuracy is high, meaning the models are well-trained and able to make accurate predictions. Comparing our estimated gCF (on blue) and the ground truth of gCF (on green), we see that our method can accurately estimate the causal fairness in the graph data. We also observe that RD and CF are quite different from gCF, indicating that one cannot simply use RD and CF to estimate gCF.

Next, we build fair node classification models on the generated graph data using the proposed method and baselines. The performance of classification prediction and fairness is shown in Tab. 2. For a fair comparison, all the models are trained to be fair based on the fairness metrics used by their own (i.e., GCN-RD uses RD and GCN-IID uses CF as their fairness metrics). As illustrated in the **Own Metric** column, all the models are well-trained and fair (for MPVA, its own metric is shown in the gCF column). Then, we present the accuracy, estimated gCF, and the ground truth gCF of all methods. As can be seen, although the baselines, including GCN-RD and GCN-IID, are considered fair based on their own fair metrics, they exhibit significant bias from *graph-based* causal fairness (i.e., gCF) perspective. In addition, the baseline methods neglect the potential effect of the graph structure while attempting to address the bias, resulting in a compromise of accuracy. On the other hand, our proposed MPVA achieves the best performance in terms of both accuracy and graph-based causal fairness gCF. We further compare our proposed MPVA with the state-of-the-art graph-based bias mitigation algorithms, FairGNN and GEAR. FairGNN aims to mitigate statistical bias in graph data, while GEAR aims to alleviate counterfactual bias in graph data. For a comprehensive comparison, we compare the trade-off between model bias and performance for MPVA,

FairGNN, and GEAR. As shown in Fig. 3, we tune each model multiple times to obtain various bias-accuracy trade-offs and plot the corresponding fairness/bias measurement used by the models and the true gCF derived from the data generation process at various accuracy levels in each subplot. We see that for our method MPVA, the estimated fairness aligns with the true causal fairness gCF at every accuracy level, thanks to our graph causal inference technique. However, for FairGNN, and GEAR, the measured fairness is significantly different from the true fairness, implying that they cannot guarantee to obtain a fair model by fine-tuning models to balance the bias-accuracy trade-off.

Table 3: Results of various methods on real-world datasets.

| Dataset | Method | Acc | RD | CF | gCF |
|---|---|---|---|---|---|
| Credit | GCN | $0.8192_{\pm 0.0005}$ | $0.0195_{\pm 0.0014}$ | $0.0049_{\pm 0.0001}$ | $0.0705_{\pm 0.0123}$ |
| | GCN-RD | $0.7988_{\pm 0.0062}$ | $0.0057_{\pm 0.0044}$ | $0.0055_{\pm 0.0001}$ | $0.0540_{\pm 0.0145}$ |
| | FairGNN | $0.7930_{\pm 0.0086}$ | $0.0047_{\pm 0.0012}$ | $0.0043_{\pm 0.0008}$ | $0.0404_{\pm 0.0251}$ |
| | GCN-IID | $0.8065_{\pm 0.0008}$ | $0.0100_{\pm 0.0023}$ | $0.0010_{\pm 0.0003}$ | $0.1360_{\pm 0.0833}$ |
| | NIFTY | $0.7933_{\pm 0.0146}$ | $0.0543_{\pm 0.0068}$ | $0.0079_{\pm 0.0006}$ | $0.0981_{\pm 0.0165}$ |
| | GEAR | $\mathbf{0.8075}_{\pm \mathbf{0.0005}}$ | $0.0260_{\pm 0.0108}$ | $0.0055_{\pm 0.0003}$ | $0.0278_{\pm 0.0111}$ |
| | FairINV | $0.7720_{\pm 0.0205}$ | $0.0111_{\pm 0.0139}$ | $0.0087_{\pm 0.0003}$ | $0.0295_{\pm 0.0226}$ |
| | MPVA | $0.8054_{\pm 0.0033}$ | $0.0142_{\pm 0.0046}$ | $0.0075_{\pm 0.0007}$ | $\mathbf{0.0036}_{\pm \mathbf{0.0033}}$ |
| German | GCN | $0.9758_{\pm 0.0027}$ | $0.0771_{\pm 0.0083}$ | $0.0704_{\pm 0.0004}$ | $0.6080_{\pm 0.2596}$ |
| | GCN-RD | $0.9608_{\pm 0.0054}$ | $0.0046_{\pm 0.0023}$ | $0.0354_{\pm 0.0005}$ | $0.2657_{\pm 0.0600}$ |
| | FairGNN | $0.8183_{\pm 0.0162}$ | $0.0051_{\pm 0.0038}$ | $0.0536_{\pm 0.0012}$ | $0.8263_{\pm 0.0526}$ |
| | GCN-IID | $0.7643_{\pm 0.0118}$ | $0.1719_{\pm 0.0133}$ | $0.0047_{\pm 0.0011}$ | $0.2994_{\pm 0.0317}$ |
| | NIFTY | $0.8113_{\pm 0.0224}$ | $0.0975_{\pm 0.0347}$ | $0.0566_{\pm 0.0011}$ | $0.9987_{\pm 0.0019}$ |
| | GEAR | $\mathbf{0.9717}_{\pm \mathbf{0.0187}}$ | $0.0689_{\pm 0.0275}$ | $0.0359_{\pm 0.0040}$ | $0.3667_{\pm 0.3769}$ |
| | FairINV | $0.8200_{\pm 0.0433}$ | $0.0428_{\pm 0.0508}$ | $0.0607_{\pm 0.0063}$ | $0.5844_{\pm 0.3669}$ |
| | MPVA | $0.9283_{\pm 0.0353}$ | $0.1136_{\pm 0.0332}$ | $0.0667_{\pm 0.0014}$ | $\mathbf{0.0030}_{\pm \mathbf{0.0032}}$ |

## 5.4 RESULTS ON REAL DATA

We further conduct extensive experiments on real-world data. We first train a naive Graph Convolutional Network (GCN) without any bias mitigation methods, run fairness-aware methods, and repeat five independent experiments. The results are shown in Table 3. As can be seen, in the original GCN, there is a big gap between gCF and CF, which implies that the IID causal metric is not accurate for measuring non-IID causal fairness in the graph. In both two datasets, the MPVA is able to **outperform** all other baselines in terms of gCF with a mild accuracy decrease compared with the classic GCN model. Although other methods achieve fairness regarding their own metrics, they fail to meet the gCF requirements. FairGNN neglects the causality-based bias, which results in compromising accuracy in order to achieve fairness. The baseline GEAR can have a good accuracy performance. However, it fails to eliminate bias more effectively. For example, in the German dataset, there is a significant bias that the GEAR is unable to completely remove. The results show that existing fairness methods cannot guarantee fairness for gCF. In summary, the results are consistent with those in the semi-synthetic datasets, demonstrating the superiority of the proposed method.

## 6 CONCLUSIONS

This paper has addressed a critical gap in fair machine learning, which traditionally relies on the Independent and Identically Distributed assumption. We focused on the graph settings where data instances are interconnected. Employing the Network Structural Causal Model (NSCM) framework, we proposed the principles of Decomposability and Graph Independence to facilitate causal inference using $do$-calculus in these non-IID settings and developed the Message Passing Variational Autoencoder for Causal Inference (MPVA) to enable the computation of interventional distributions. Empirical evaluations on semi-synthetic and real datasets have shown that MPVA surpasses baseline methods by more accurately approximating interventional distributions and reducing bias.

## 7 REPRODUCIBILITY STATEMENT

To ensure reproducibility of our work, we provide comprehensive implementation details and experimental setup information in the appendix. The complete source code for MPVA, including the Message Passing Variational Autoencoder architecture and training procedures, is attached as a supplementary material and will be made available upon acceptance. All hyperparameter settings, network architectures, and optimization details are specified in Section D of the appendix. The synthetic data generation process following the Network Structural Causal Model is fully described as well in Section D. For real-world datasets (Credit and German), we provide detailed preprocessing steps and train/validation/test splits in the supplementary materials. All baseline implementations follow their original papers with publicly available code, and we report means and standard deviations across five independent runs to ensure statistical reliability. The theoretical foundations, including proofs of our identifiability conditions (Decomposability and Graph Independence), are provided in the appendix to enable verification of our theoretical contributions.

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

## A  RELATED WORK

**Graph-based Causal Inference**. Recently, the IID assumption in causal inference has been investigated and extended. A set of works extends the graphical causal modeling framework. Ogburn & VanderWeele (2014) extended DAGs for the interference relationships among individuals. Sherman & Shpitser (2018) modeled interference using chain graphs that permit modeling unknown interactions between individuals. Bhattacharya et al. (2019) proposed an interventional method for estimating causal effects under data dependence when the structure. In addition to the graphical modeling, researchers defined various effects to capture the relationships among variables and data points. Shpitser & Pearl (2008) defined the individual and group average direct and indirect effects (a.k.a. spillover effect) in the interference situations. Ogburn & VanderWeele (2014) developed direct interference, interference by contagion and infectiousness, and avocational interference. In additional modeling, recent years have witnessed a rise in papers on causal interference effect estimation. Hudgens & Halloran (2008) and VanderWeele & Tchetgen Tchetgen (2011) have developed new randomized procedures for unbiased estimands. Fatemi & Zheleva (2020) proposed a new experiment design approach to minimize interference bias and selection bias during estimation. Tchetgen Tchetgen et al. (2021) proposed a general g-computation method for causal interference.

**Fairness on Graphs**. Recently, algorithmic bias in machine learning has garnered significant attention from the research community, spawning a proliferation of methods and studies (Binns, 2020; Jiang et al., 2023; Verma & Rubin, 2018). Various notions of fairness have been proposed to define fairness formally, which can be categorized into two groups. The first category is *statistical parity*, which means the proportions of receiving favorable decisions for the protected and non-protected groups should be similar. The quantitative metrics derived from *statistical parity* include *risk difference*, *risk ratio*, *relative change*, and *odds ratio* (Wu et al., 2019; Hardt et al., 2016; Pedreschi et al., 2012). Recent works (Agarwal et al., 2021; Bose & Hamilton, 2019; Buyl & Bie, 2020; Dai & Wang,

2021; Dong et al., 2021; Kang et al., 2020) mitigate bias in node representation learning. Most of the works (Beutel et al., 2017; Zhang et al., 2018) are focused on adversarial learning, ensuring that the learned representations do not reliably predict the associated sensitive attribute. These works focus on eliminating the statistical dependency between the sensitive attribute and prediction from the learned representation but neglect the bias raised by the feature or graph structure due to the causal effect. Another category of fairness notion is counterfactual fairness, which is developed mainly under the structural causal model (SCMs) (Pearl, 2009). There are a few works (Ma et al., 2022; Agarwal et al., 2021; Yang et al., 2024) that extend the counterfactual fairness to graphs. However, most of these works ignore the potential biases introduced by the sensitive attributes of neighboring nodes and the causal effect of sensitive attributes on other nodes.

## B    DISCUSSION OF PROPOSED CONDITIONS

### B.1    DECOMPOSABILITY

The Decomposability condition states that when a node's outcome is influenced by both its own attributes and its neighbors' attributes, the influence can be decomposed into two steps: first aggregating information from neighbors, then combining it with the node's own attributes. This condition helps make causal inference tractable in graph settings by providing a structured way to model how information flows through the network. With this condition, we can decompose the causal effect of the neighbors into the message passing mechanism, denoted as $f^{MP}(\cdot)$, and the internal causal mechanism within the node, denoted as $f^{INT}(\cdot)$.

For example, consider a social media platform that uses an algorithm to deliver purchase discounts to users. We posit that a user may choose to subscribe to the supplier (i.e., $X$), and if he/she decides to subscribe to the supplier, the chance of receiving the discount will increase. Due to the connections in the social network, we posit that a user's decision to subscribe is influenced by his/her own situation (i.e., $S$) as well as his/her neighbors. Then, Decomposition posits that the user will first aggregate the situations from all the neighbors and then combine them with his/her own situation when making the decision. This two-step process allows us to separately model the network effects and individual effects while still capturing their joint influence on the final outcome.

### B.2    GRAPH INDEPENDENCE

The Graph Independence condition posits that the graph structure (captured by the node color $C$) is independent of the exogenous variable (denoted as $U_X$), which ensures that all relevant information from the neighborhood regarding the interventional value of $X$ can be effectively summarized in the intermediate variable $A$. As a result, the implication of Proposition 2 and Condition 2 together implies that we can convert the networked causal diagram in Fig. 1c (from the main paper) to an equivalent causal diagram as shown in Fig. 3 (from the main paper).

In the example of a social network where users are connected based on shared interests or demographics, the Graph Independence condition means that the network connections themselves (who is friends with whom) are not influenced by exogenous factors that also affect the non-sensitive attributes. This condition allows us to treat the network structure as fixed and unaffected by interventions, thereby preventing cyclic dependencies in the causal diagram (i.e., Fig 3). In other words, when we intervene on a node's attributes, we posit this intervention does not alter the underlying network topology.

## C    PROOF OF THEOREM 3 IN THE MAIN PAPER

For any two variables $X, Y$ in an SCM, let $y(u)$ be the value of $Y$ of an instance whose exogenous variable is $u$, and $y_x(u)$ be the value of $Y$ under the intervention $x(u) = x$. According to Pearl (2009), we have the following lemmas.

**Lemma 5.** *Given the causal diagram in Fig. 6 in the main paper, we have that $X_a \perp A$ for any $a$.*

**Lemma 6.** *Given the causal diagram in Fig. 6 in the main paper, for any node $u$, we have that $x_{\mathsf{s}}(u) = x_{\mathsf{s},c,a}(u) = x_a(u)$ if $a_{\mathsf{s}}(u) = a$ and $c(u) = c$.*

**Lemma 7.** *Given the causal diagram in Fig. 6 in the main paper, for any node $u$, we have that $a_{\mathsf{s}}(u) = a_{\mathsf{s},c}(u)$ if $c(u) = c$.*

**Theorem 3.** *Given the causal diagram in Fig. 6 in the main paper, we have*

$$P(x|do(\mathsf{s}) = s) = \sum_c P(c) \sum_a P(x|a)P(g(\mathsf{s}, c) = a).$$

*Proof.* According to the formula of the conditional probability, we directly have

$$P(x|do(\mathsf{s}) = s) = \sum_{c,a} P(x|c, a, do(\mathsf{s}) = s)P(c, a|do(\mathsf{s}) = s)$$

$$= \sum_{c,a} P(x|c, a, do(\mathsf{s}) = s)P(c|do(\mathsf{s}) = s)P(a|c, do(\mathsf{s}) = s).$$

Since $\mathsf{S}$ is not a descendent of $C$ in the causal diagram, it follows that

$$P(x|do(\mathsf{s}) = s)$$
$$= \sum_{c,a} P(x|c, a, do(\mathsf{s}) = s)P(c)P(a|c, do(\mathsf{s}) = s).$$

According to Lemma c, we have $P(a|c, do(\mathsf{s}) = s) = P(a|do(c), do(\mathsf{s}) = s)$ which can be rewritten as $P(g(\mathsf{s}, c) = a)$ below using the mapping $g$. By similarly applying Lemma b, we have $P(x|c, a, do(\mathsf{s}) = s) = P(x|do(c), do(a), do(\mathsf{s}) = s) = P(x|do(a))$. As a result, we have

$$P(x|do(\mathsf{s}) = s)$$
$$= \sum_{c,a} P(x|do(a))P(c)P(g(\mathsf{s}, c) = a).$$

Then, we rewrite the above equation as

$$P(x|do(\mathsf{s}) = s)$$
$$= \sum_{c,a} \sum_{a'} P(x|a', do(a))P(a')P(c)P(g(\mathsf{s}, c) = a)$$

According to Lemma a, we have $P(x|a', do(a)) = P(x|a, do(a))$, which is equal to $P(x|a)$ according to the Composition Axiom. Finally, we have that

$$P(x|do(\mathsf{s}) = s) = \sum_{c,a} P(x|a) \sum_{a'} P(a')P(c)P(g(\mathsf{s}, c) = a)$$

$$= \sum_c P(c) \sum_a P(x|a)P(g(\mathsf{s}, c) = a).$$

Hence, the theorem is proved. □

## D  EXPERIMENTS

### D.1  DATASETS

In the semi-synthetic dataset, we leverage the Credit Dataset(Yeh, 2016). To get full control over the data generation, we define the data generation mechanism as follows. Denoting the original features in the dataset as $C$, we first build a classifier to predict $S$. We use this classifier to soft label the prediction of $S$ to get the probability distribution of generating $S$. Then, we similarly build another classifier to estimate $Y$ in order to get the probability distribution of generating $Y$. After that, we

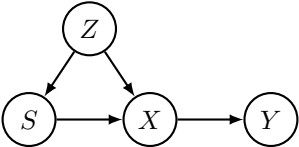

Figure 4: The networked causal diagram for node classification.

randomly initialize a GNN $g(\cdot) : \mathcal{S} \to \mathcal{A}$ to mimic the influence of neighbors' sensitive attributes of a certain node on its own attributes. Finally, we generate our semi-synthetic dataset as follows:

$$S_i^g \sim Bernoulli(p), X_i^g = g(S^g) + C_i + \xi$$

$$Y_i^g = f_y(X_i^g)$$

where the sensitive attribute is sampled from a Bernoulli distribution, $p = f_s(C_i)$ is the probability of $S_i^g = 1$, and $\xi$ is the random noise that is sampled from Gaussian distribution. We simulate the probability of each edge $(i, j)$ based on the similarity between $X_i^g$ and $X_j^g$. We generate ground truth counterfactual data by setting all $S^g$ to 1 and 0 in order to get positive and negative intervened distribution, respectively.

As for the real-world graphs, we conduct experiments on widely used real-world datasets, namely Credit Defaulter (Yeh, 2016) and German (Hofmann, 1994). The details of the datasets are as follows.

**Credit Defaulter**: the nodes in the dataset are used to represent the credit card users, and the edges are formed based on the similarity of the payment information. The task is to classify the default payment method with the sensitive attribute "Sex". We treat "Education", "Marriage", and "Age" as $Z$, i.e., variables other than $S$ that directly affect $X$.

**German**: The German credit network consists of 1,000 nodes, which represent clients of a German bank. These nodes are interconnected based on the similarity of their credit accounts. The objective is to categorize clients as either excellent or bad credit risks, taking into consideration the clients' gender as the sensitive attribute. We treat "YearsAtCurrentJob" and "JobClassIsSkilled" as $Z$.

### D.2   IMPLEMENTATION

We use a one-layer message-passing neural network to aggregate the sensitive causal effect of one-hop neighbors. We train MPNN with 0.01 learning rate and 500 epochs. The models are all implemented using Pytorch 1.12.0 and PyG 2.4.0 and evaluated in a Linux server with an Intel(R) Core(TM) 19-10900X CPU and an NVIDIA GeForce RTX 3070 GPU. The memory consumption is about 2000 MiB. We use cVAE to reconstruct features conditional on $\hat{a}$ and $c$. For training cVAE, the learning rate is 0.01, the epochs is 800. Experimental results are averaged over five repeated executions. We use the Adam optimizer for both two components of our proposed framework and implement our method with Pytorch. For the constraint-based method, we train a multilayer perceptron (MLP) with corresponding fairness regularization terms to achieve fairness.

**Risk difference on IID data:** The risk difference usually refers to the difference of the positive predictions between the favorable group and the non-favorable group. It is easy to compute the possibility of output given certain sensitive attribute $P(y \mid s) = \mathbb{E}_{x|s}P(y \mid x)$. Then, we design the regularization term as $P(y \mid s^+) - P(y \mid s^-)$.

**Causal inference on IID data:** For IID data, we usually use a structural causal model to describe the causal relationship between two variables. For example, the causal relationship between $S$ and $X$ is given by:

$$x = f(s, u).$$

 We consider the same causal structure in Fig. 4 but neglect the network causal effect. To compute the possibility of output given certain intervention on the sensitive attribute $P(y|do(s))$:

$$P(y \mid do(s)) = \sum_{z,x} P(z)P(x \mid s,z)P(y \mid x)$$

$$= \sum_{z,x} P(z \mid s)\frac{P(s)}{P(s \mid z)}P(x \mid s,z)P(y \mid x)$$

$$= \sum_{z,x} P(z,x \mid s)\frac{P(s)}{P(s \mid z)}P(y \mid x)$$

$$= \mathbb{E}_{z,x \sim P(z,x|s)}\left[\frac{P(s)}{P(s \mid z)}P(y|x)\right] \tag{4}$$

Then, we design the regularization term as $P(y \mid do(s^+)) - P(y \mid do(s^-))$.

### D.3 SENSITIVITY ANALYSIS

#### D.3.1 MULTI-HOP CAUSAL EFFECT ANALYSIS

We further demonstrate the proposed MPVA framework is capable of capturing multiple-hop causal effects in graph data. We generate the influence of neighbors' sensitive attributes of a certain node within the range of three-hop. As shown in Fig. 5, when the MPNN module has the same number of layers as the neighborhood hops of the generating model (which is 3 in Fig. 5), it can achieve the best performance of estimating the interventional distribution. This may be because of underfitting when the number of layers of MPNN is too small, and overfitting when the number of layers is too large. We also observe that the variance of the results is minimal when the layer number is the same as the neighborhood hops. This can be used to help us pick an appropriate number of layers in practice.

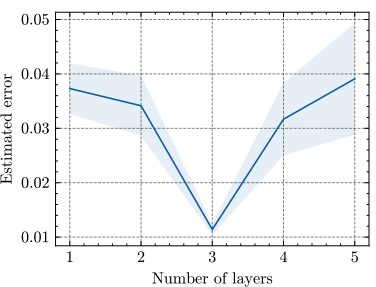

Figure 5: The impact of the number of MPNN layers against estimated error.

Table 4: gCF Difference under different dependency strengths.

| Violation Strength | gCF Difference |
|---|---|
| 0.0 | $0.0075_{\pm 0.0007}$ |
| 0.5 | $0.0592_{\pm 0.0012}$ |
| 1.0 | $0.1142_{\pm 0.0016}$ |

Table 5: Runtime and memory usage comparison of different methods.

| Method | Time (s) | Memory (MB) |
|---|---|---|
| FairGNN | 81.7968 | 3639 |
| NIFTY | 17.0893 | 2419 |
| GEAR | $> 3600$ | – |
| FairINV | 63.7923 | 2103 |
| MPVA | 29.2358 | 1365 |

#### D.3.2 SIMULATION OF VIOLATION OF PROPOSED CONDITIONS

Additionally, we simulate a scenario using the synthetic dataset where the proposed conditions are violated. To facilitate the analysis, we introduce a dependency between $U_X$ and $A$ by incorporating a weighted multiplicative interaction (e.g., $c \cdot A \cdot U_X$) during the data generation process where a coefficient ($c$) is introduced to control the violation or dependency strength. The we measure the absolute difference between our estimated gCF and the True-gCF. As shown in Tab. 4, although both methods minimize their respective fairness metrics, they fail to achieve graph-based causal fairness, as reflected in the gCF and True-gCF columns, which are consistent with the observations reported in our paper.

### D.4 COMPUTATIONAL COST

To compare the computational cost of our method and the baselines, we measured the runtime and memory usage for our method and the baselines on the Credit dataset. As shown in Tab. 5, MPVA achieves the lowest memory usage (1365 MB) and offers the second fastest running time (29.2 seconds), compared to the fastest runtime (17.0893 seconds) of NIFTY, demonstrating its computational efficiency. The superior memory efficiency of MPVA makes it particularly suitable for large-scale graph applications where memory constraints are critical. Notably, GEAR exceeded the 3600-second time limit and could not complete the experiment, highlighting the scalability challenges of existing methods.

## E LLM USAGE DISCLOSURE

In the preparation of this manuscript, Large Language Models, specifically ChatGPT and Claude, were utilized to enhance the clarity and coherence of the written text. All content generated by these models has been thoroughly reviewed, fact-checked, and validated by the authors to ensure accuracy and adherence to the research findings and methodological rigor presented in this work.

## F ETHICS STATEMENT

The developed method has significant potential to combat bias in the increasingly ubiquitous graph data in numerous fields, such as social networks, bioinformatics, and advertisement. We acknowledge that practitioners should remain vigilant about potential unintended consequences. For instance, our approach could help reduce discriminatory outcomes in social networks, but care must be taken to ensure that the chosen sensitive attributes and fairness metrics align with context-specific ethical considerations and legal requirements. We encourage future work to explore the long-term societal impacts of deploying such systems at scale, including effects on privacy, trustworthy, and social dynamics.

