# OpenReview forum: "Causally Fair Node Classification on Non-IID Graph Data"
_ICLR.cc/2026/Conference — ICLR 2026 Conference Withdrawn Submission_

### Official Review · Reviewer_Tsn9 · 2025-10-26

**Soundness:** 3
**Presentation:** 3
**Contribution:** 2
**Rating:** 4
**Confidence:** 3

**Summary:**

In this paper, the authors introduce a causality-based fairness framework for graph machine learning that removes the IID assumption common in existing methods. Using the network structural causal model, it defines two theoretical conditions, decomposability and graph independence, to enable causal inference with do-calculus on non-IID data. The proposed message passing variational autoencode estimates interventional distributions and integrates them into fair node classification. Experiments on semi-synthetic and real datasets show improved causal estimation and fairness performance over standard approaches.

**Strengths:**

- The paper’s main theoretical contribution is indeed the extension of causal inference from the IID assumption to graph-structured, networked data via the network structural causal model.

- The integration of message passing neural networks with a variational autoencoder provides a coherent deep learning implementation for estimating interventional distributions.

- Results show that the proposed MPVA performs better than the baseline fair GNNs included.

**Weaknesses:**

- The decomposability and graph independence conditions are unlikely to hold in most real-world graphs. Real graphs often exhibit dependencies between the node structure and exogenous variables (e.g., social homophily), limiting the scope of the theoretical framework.

- Although the model claims to estimate interventional distributions, it does not analyze what these causal effects reveal about bias sources in the data

- It does not thoroughly discuss computational complexity or the scalability of MPVA on large graph datasets.

- Both the Credit and German datasets are inherently tabular and not naturally graph-structured.

**Questions:**

Please check the weaknesses.

---

### Official Review · Reviewer_qn27 · 2025-10-29

**Soundness:** 3
**Presentation:** 2
**Contribution:** 3
**Rating:** 4
**Confidence:** 2

**Summary:**

This paper addresses the challenge of ensuring fairness in graph-based learning, where the standard i.i.d. assumption no longer holds due to the network interconnections. Building on the Network Structural Causal Model (NSCM), the authors introduce the principles of Decomposability and Graph Independence to enable causal inference via do-calculus in non-IID settings. They further develop the Message Passing Variational Autoencoder (MPVA) to estimate interventional distributions and incorporate them into a fairness-regularized training framewok. Experiments on semi-synthetic and real-world datasets show that MPVA better approximates interventional distributions and achieves lower bias than existing baselines.

**Strengths:**

1. Novelty: The paper tackles a novel and underexplored problem: defining a causal fairness notion for non-IID (graph-structured) data. This direction is timely and relevant, as traditional fairness definitions based on the i.i.d. assumption fail in networked settings.
2. Significance: The proposed framework has strong potential impact, since non-IID fairness is essential for a variety of graph-based learning applications, such as social network analysis and credit risk prediction.
3. Quality: The experimental section is thorough, covering both semi-synthetic and real-world datasets. Results consistently demonstrate the superiority and robustness of the proposed method compared to state-of-the-art baselines.

**Weaknesses:**

1. Some notations and statements could be better explained. The model figure is not very informative, and a clearer diagram would help improve readability.
2. The paper would benefit from additional illustration or intuitive explanation of the theoretical assumptions (e.g. Decomposability and Graph Independence) to make their implications more accessible.
3. It remains unclear whether the proposed MPVA framework scales to large graphs or how sensitive its performance is to the accuracy of the estimated interventional distributions.

While I find the causal fairness framing and theoretical analysis interesting, I am not deeply familiar with graph-based node classification. My evaluation mainly focuses on the causal assumptions, clarity, and experimental methodology rather than task-specific design details.

**Questions:**

1. I find the meaning of the double-line arrows in the networked causal diagrams unclear. For instance, if there is a double line between $S\Rightarrow X$, does it imply dependence among instances of $S$, among instances of $X$, or cross-node dependence between $S$ and $X$? The semantics of this notation seem ambiguous to me.
2. In Figure 1(f), the causal graph seems to suggest that the double line indicates dependencies only among the variable $S$. If that is the intended interpretation, I am curious why a solid edge $C\rightarrow A$ is also introduced. This edge implies $C\rightarrow X$, while the connection between $C$ and $S$ is dashed. Why can not represent it as $S\leftarrow C\leftarrow X$? If cyclic graphs are disallowed in your formulation, it would be clearer to state this explicitly.
3. Regarding the Decomposability condition, the paper gives a particular functional decomposition of the structural equations. Could you please clarify whether this decomposition is meant to be general, or only holds for a specific class of models? It would be helpful if you could provide concrete examples where such a nested structure can be satisfied.

---

### Official Review · Reviewer_PnJ6 · 2025-10-31

**Soundness:** 2
**Presentation:** 2
**Contribution:** 2
**Rating:** 2
**Confidence:** 4

**Summary:**

This paper proposes MPVA, a causality-based framework for fair node classification on graphs that addresses the limitation of existing fair ML methods which assume IID data. The authors establish two conditions enabling causal inference via do-calculus in non-IID settings, then implement this using message-passing neural networks with conditional variational autoencoders to estimate interventional distributions and enforce fairness through regularization.

**Strengths:**

1.	The paper tackles the underexplored problem of causal fairness in non-IID graph settings.

2.	By focusing on interventional distributions rather than counterfactuals, the method sidesteps well-known identifiability problems in counterfactual inference.

**Weaknesses:**

1.	While the paper establishes theoretical conditions for when $do$-calculus applies, the experiments do not systematically verify that these conditions hold for the datasets used.
2.	The paper assumes a specific causal structure where the sensitive attribute $S$ affects node features $X$ through an intermediate variable $A$, which then affects the outcome $Y$ (i.e., $S \rightarrow A \rightarrow X \rightarrow Y$). However, this structure fails to account for direct causal relationships that commonly exist in practice. For example, gender (a sensitive attribute) can directly affect height (a feature) without any intermediate variable.
3.      More general, is the causal model introduced appropriate for the proposed task? According to Pearl’s causal hierarchy, the first rung, i.e., Association, does not require any causal information.
4.	Only two real-world datasets (Credit Defaulter and German) are used, both from the financial domain. This is insufficient to demonstrate generalizability across different types of graphs (social networks, citation networks, biological networks, etc.) or different application domains.
5.	Insufficient discussion and distinction from recent and relevant fair graph learning works [1,2].
[1] Wang, Zichong, et al. "Fair graph u-net: A fair graph learning framework integrating group and individual awareness." proceedings of the AAAI conference on artificial intelligence. Vol. 39. No. 27. 2025.
[2] Zhu, Yuchang, et al. "Fair graph representation learning via sensitive attribute disentanglement." Proceedings of the ACM Web Conference 2024. 2024.
6.      It is recommended to enhance the paper's organization by incorporating more subsections, theorems, and similar structural elements.

**Questions:**

See above, also, could the authors explain how MPVA handles cases where S directly affects X?

---

### Official Review · Reviewer_biV2 · 2025-11-06

**Soundness:** 3
**Presentation:** 3
**Contribution:** 3
**Rating:** 6
**Confidence:** 3

**Summary:**

This paper tackles causality-based fairness for node classification when data are *non‑IID* and connected as a graph. It starts from the Network Structural Causal Model (NSCM) and introduces two conditions---**Decomposability** (separating neighborhood aggregation from within‑node mechanism) and **Graph Independence** (node-structure summary independent of exogenous noise)---under which do‑calculus can be applied to network data. Using Weisfeiler–Lehman (WL) "node colors" to summarize local structure, the authors show that the networked diagram can be transformed into an equivalent DAG and derive a computable formula for the interventional distribution (Theorem 3). Crucially, they argue $ P(x\mid do(s)) $ can be approximated with the *same* model used to fit $P(x)$ by "intervening" on (S) in the learned message‑passing mechanism (Corollary 4).

They instantiate this as **MPVA** (message passing variational autoencoder): an MPNN learns a representation of aggregated neighbor effects; an MLP/cVAE learns $P(x\mid a,z)$; and at inference they perform *abduction-action-prediction* by re‑computing $ \tilde a $ under $do(s^{+})$ or $do(s^{-})$ and decoding interventional $ \tilde x $. A fairness regularizer is then the interventional prediction gap using a differentiable surrogate for the indicator, which is combined with the usual classification loss.

**Strengths:**

1. Identifies two general conditions (decomposability, graph independence) under which *do‑calculus* extends to non‑IID graphs, yielding a *closed‑form interventional expression* (Theorem 3). This is a nice result IMO.
 2. Uses WL colors to ground the structural summarization that enables the graph isomorphism logic with causal identification.
3. Semi‑synthetic experiments include ground‑truth interventions and show close agreement. Reasonable sensitivity analysis and assumption‑violation study.
4. I found the paper overall clear. Diagrams and equations are well integrated. The training/inference details are explicit.
5. The authors demonstrate that *IID causal fairness (CF) and statistical parity (RD)* can *severely disagree* with *graph‑causal fairness (gCF)* (Tables 1 and 3), which might have practical importance for fairness in some settings.

**Weaknesses:**

1. **Graph Independence** (exogenous $U_X$ independent of structure C) is unlikely in many social graphs with homophily or formation dynamics. The paper shows degradation under violations (Table 4) but does not provide diagnostics or robustness strategies on real data. Does it make sense to add stress tests on real graphs (e.g. degree- or community‑stratified analyses)?

2. **Scope of interference patterns.** Most empirical work assumes interference flows only from S to X (Fig. 1e), whereas real tasks may have interference on (Y) directly or multi‑edge spillovers. Maybe consider providing an experiment where $S \to Y$ or $X \leftrightarrow X$ interference exists, and verify that MPVA still estimates $P(x\mid do(s))$ or $P(y\mid do(s))$ appropriately (or clarify how to modify the architecture for that case).

3. It's not fully clear how much the cVAE contributes beyond an MPNN‑regressor; an ablation without cVAE or with different decoders would help. Clarify the role of WL colors in implementation (are colors explicitly computed, or is structure entirely learned by the MPNN?).

**Questions:**

1.  For Table 3, could you *recompute gCF* with an **external estimator** (even if less accurate), to reduce the risk of "judge = participant"? Alternatively, maybe have a "frozen MPVA‑measurer" trained without access to each model's outputs, then evaluate all models with that fixed measurer?

3. Unless I am misunderstanding, your fairness penalty uses a global $do(s)$. How would the method change for a per‑node intervention $do(s_i)$ while neighbors remain at observed values, i.e., an individual‑level gCF? Would Theorem 3 or Corollary 4 extend?

4. In MPVA, do you explicitly compute WL colors $C$ or is structural context implicit in the MPNN? If implicit, how do you reconcile Proposition 2's dependence on $S,C$ with learned embeddings?

5. Can you provide at least one experiment (synthetic is fine) where interference also affects $Y$ directly (Fig. 1e extended), and show how $P(y\mid do(s))$ is estimated?

6. What fails if we replace cVAE with a simpler conditional regressor? Is the variational layer primarily for uncertainty calibration (to enable expectation over $x$)?

---

### Note · Authors · 2025-11-23

I have read and agree with the venue's withdrawal policy on behalf of myself and my co-authors.